# Effects of Inter- and Intra-Specific Interactions on Moose Habitat Selection Limited by Temperature

**Heng Bao** [1,2,†], **Penghui Zhai** [1,2,†], **Dusu Wen** [1,2], **Weihua Zhang** [3], **Ye Li** [3], **Feifei Yang** [1,2], **Xin Liang** [1,2], **Fan Yang** [1,2], **Nathan J. Roberts** [1,2], **Yanchun Xu** [1,2] and **Guangshun Jiang** [1,2,*]

1   Feline Research Center of National Forestry and Grassland Administration, College of Wildlife and Protected Area, Northeast Forestry University, Harbin 150040, China
2   Research Center for Northeast Asia Biodiversity Conservation, Harbin 150040, China
3   Inner Mongolia Hanma National Nature Reserve, Genhe, Hulunbuir 022359, China
*   Correspondence: jiangguangshun@nefu.edu.cn
†   These authors contributed equally to this work.

**Abstract:** Habitat selection and daily activity patterns of large herbivores might be affected by inter- and intra-specific interaction, changes of spatial scale, and seasonal temperature. To reveal what factors were driving the habitat selection of moose, we collected moose (*Alces alces*) and roe deer (*Capreolus pygargus bedfordi*) occurrence data, analyzed the multi-scale habitat selection and daily activity patterns of moose, and quantified the effects of spatial heterogeneity distribution of temperature, as well as the occurrence of roe deer on these habitat selection processes. Our results suggested that moose and roe deer distribution spatially overlap and that moose habitat selection is especially sensitive to landscape variables at large scales. We also found that the activity patterns of both sexes of moose had a degree of temporal separation with roe deer. In the snow-free season, temperatures drove moose habitat selection to be limited by threshold temperatures of 17 °C; in the snowy season, there were no similar temperature driving patterns, due to the severe cold environment. The daily activity patterns of moose showed seasonal change, and were more active at dawn and nightfall to avoid heat pressure during the snow-free season, but more active in the daytime for cold adaptation to the snow season. Consequently, this study provides new insights on how the comprehensive effects of environmental change and inter- and intra- specific relationships influence the habitat selection and daily activity patterns of moose and other heat sensitive animals with global warming.

**Keywords:** camera trapping; habitat selection; inter and intra-specific relationships; spatial scale; temperature

## 1. Introduction

Habitat selection is one of the basic contents of wildlife ecology, which can indicate spatial and temporal population dynamics [1], closely relates to the survival and reproduction of species [2], and playd an important role in the protection, management, and recovery of endangered species [3,4]. Habitat selection is a complex process affected by many ecological factors, including resource availability, shelter condition, predation risk, and intra-specific competition.

Recently, many studies have clarified the importance of spatial scale on selection processes. They emphasize that scale selection during analyses might lead to different conclusions, and that the role and significance of single habitat factors can vary with spatial scale [5–7]. When compared with single-scale and non-optimal models, the predictive ability of multi-scale optimal models is much improved [6–8]. Therefore, determining and utilizing the optimal scale of habitat variables in habitat selection models has attracted more and more attention from ecologists.

Seasonal changes, such as temperature, food availability, and ecosystem structure and function, cause fluctuations in any given animal's physiological and nutritional demands and, hence, vary habitat selection rules with each season. Many ungulates seasonally migrate due to the changes in food quantity and quality with distinct seasons [9]. Moose and roe deer (*Capreolus pygargus bedfordi*) are sympatric species distributed in the Greater Khingan mountains in China, and these two species exhibit a strong competitive feeding relationship [10]. Furthermore, in regards to scale, the feeding points of moose and roe deer are different at landscape, patch, and microhabitat scales [11]. Roe deer have often been regarded as an important factor in inter-specific competition and moose habitat selection in ecological research. In addition, female and male deer show different habitat selection and space use patterns during most of the year [12]. Sexual dimorphism is widely expressed in ruminants [13]. In moose (*Alces alces*), female and male moose have different physiological characteristics, which result in behavioral variation, especially expressed in relation to environmental changes throughout much of the year [14–17].

The moose is a large herbivore, a typical circumpolar species, and might have similar cold adaptation strategies through genetic mechanisms as reindeer (*Rangifer tarandus*), polar bears (*Ursus maritimus*), and penguins (*Pygoscelis adeliae*) [18,19]. Moose have a very low tolerance to heat, especially in summer, which limits the spatial distribution boundary of their southern edge, but they are well adapted to cold environments through metabolic adaptation [20]. Seasonally, moose body temperature is affected by fat reserves, while daily variation can be attributed to environmental conditions [21]. When the summer temperature exceeds 20 °C, moose prefer to select habitats with higher forest stand and denser forest canopy, and habitat selection behavior is significantly different to that in low temperature environments [22]. Related research has confirmed that moose populations are sensitive to climate change, and are especially affected by late spring temperatures [23,24].

Despite these existing published studies on the comprehensive effects of scale, the effects of inter- and intra-specific relationships and temperature on moose seasonal habitat selection can be further improved. In this study, we tested three specific hypotheses in the snow-free season (from early April to the end of September) and snowy season (from early October to the end of March): (1) the habitat selection of moose exhibit spatial overlaps with roe deer, but varies with spatial scales, season, and the sex of the moose; (2) moose daily activity patterns and temporal overlap with roe deer vary with season and the sex of the moose; (3) moose exhibit different scalar responses to environmental temperatures, especially in the warmer snow-free season, and avoid the habitats in which the high temperatures are greater than their level of tolerance.

## 2. Methods

### 2.1. Data Collection

Our research was conducted in Hanma National Nature Reserve (51°20′02″–51°49′48″ N, 122°22′48″–122°52′46″ E), in the Greater Khingan Mountains of Inner Mongolia, which is one of the most primitive boreal coniferous forests with very little human disturbance. Hanma is located in the cold region of China, where the lowest extreme temperature reaches −58 °C. To satisfy uniform sampling, the survey area was divided into grids of 2 km × 2 km using the fishnet tool in ArcGIS 10.6 (Environmental Systems Research Institute, Inc. Beijing, China), which referred to the average home range size ($13.7 \pm 2.2$ km$^2$) of the female moose [25].

In total, we established 120 camera trapping points (Figure 1) from July 2016 to August 2017, and divided the camera photo and video data into two parts: the snow-free period (from early April to the end of September) and the snow period (from early October to the end of March). The sex of the moose was confirmed according to whether they had horns or whether the horn positions on the head were obviously raised. Statistics were not made if there were no sex features. We then counted the frequency of occurrence of male and female moose and roe deer for each camera point in the two periods; the same

species photos or videos captured within 30 min at the same camera point were recorded as independent photos or videos.

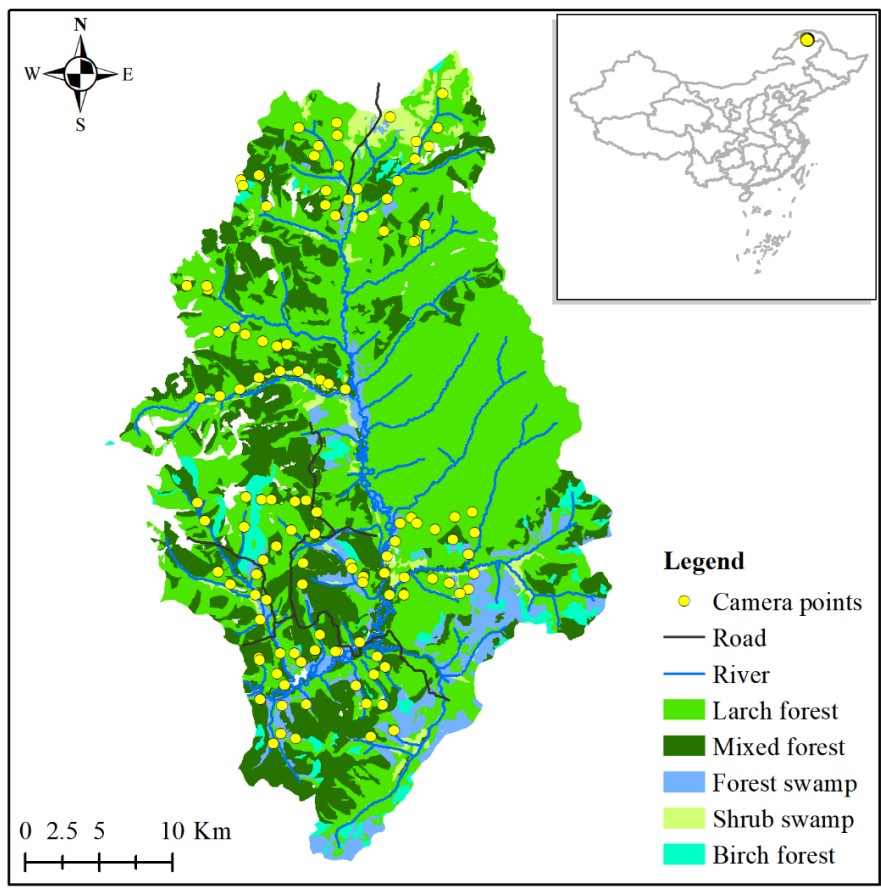

**Figure 1.** Camera trap points in Hanma National Nature Reserve, Inner Mongolia, China.

We obtained data on habitat factors in the study area from the Forest Resources Planning and Design Survey (Class II survey) in 2016. We extracted 16 habitat variables at 50 m, 100 m, 200 m, 400 m, 800 m, 1.6 km and 3.2 km spatial scales, and they included topographic factor (elevation, slope, aspect), canopy density (reflecting the effects of forest shade on animals), vegetation type (distance to larch (*Larix* spp.) forest patch, distance to birch (*Betula* spp.) forest patch, distance to mixed forest patch, distance to shrub swamp patch, distance to forest swamp patch, distance to river (reflecting the ecological function of river resources), distance to road (patrol line) (reflecting the road disturbance on animals), amount of fallen logs (which hinder the animal's ability to move around, but also provide a rich food resource), and food resources (abundance of *Betula exilis* shrub, abundance of *Rhododendron* spp. shrub, abundance of *Chosenia arbutifolia* shrub, and herbage coverage) (Table S1). The minimum scale was based on the grid size of the selected habitat factors and the available computing power. The maximum scale reflected the size of the camera grid in the study area, which was close to the average home range size of female moose ($13.7 \pm 2.2$ km$^2$) [25].

### 2.2. Multi-Scale Habitat Selection Modeling for Moose and Roe Deer

The optimal scale at which each single habitat variable affects the distribution of the moose population might be different, and the univariate optimal scale of roe deer may be different from that of moose. We therefore used generalized linear modelling (GLM) in R 4.1.0 software to establish single habitat variable selection models of moose and roe deer occurrence (Binomial) at seven scales: 50 m, 100 m, 200 m, 400 m, 800 m, 1.6 km and 3.2 km.

To determine the respective univariate optimal scales of each variable, we referred to the smallest sample Akaike Information Criterion (AICc) value [26].

We followed two principles for the elimination and selection refinement of habitat variables. Firstly, in order to eliminate the multicollinearity problem established by the model and consider the correlation of various habitat factors (Rs > 0.5), the correlation matrix of each habitat variable was obtained by using the Band Collection Statistics (BCS) tool in ArcGIS 10.6, and the variable with lower ecological explanatory power was eliminated. Secondly, we eliminated habitat variables for which $p > 0.05$ in the respective univariate model. The following correlations were detected: elevation with distance to forest swamp, distance to river, and abundance of *C. arbutifolia* shrub; distance to road with distance to shrub swamp patch and distance to mixed forest; and abundance of *B. exilis* shrub with abundance of *C. arbutifolia* shrub. In addition, distance to shrub swamp did not have a significant effect on moose or roe deer occurrence ($p > 0.05$) in the univariate models. Ultimately, we finalized a selection of 11 variables in the multi-scale habitat variable model, including elevation, slope, aspect, canopy density, herb coverage, abundance of *B. exilis* shrub, abundance of *Rhododendron* spp. shrub, amount of fallen logs, distance to birch forest, distance to larch forest, and distance to mixed forest.

We used GLM with Poisson distribution to establish the multi-scale habitat selection model for moose, roe deer, female moose, and male moose in snow and snow-free periods, and used the "Step" command to screen the optimal model of each dependent variable by AIC minimum principle in R software. In order to judge and inspect the ability of the optimal model, as well as determine the importance of habitat variables included in the optimal model, we used the piecewise Structural Equation Model (SEM) package in R software to calculate the deviation interpretation rate ($R^2$) of each optimal model and each habitat variable in the optimal model. The univariate optimal scale selection, habitat variable elimination, and habitat selection model establishment mainly followed McGarigal et al. (2016) and Macdonald et al. (2018) [7,8].

### 2.3. Spatial and Temporal Overlap for Moose and Roe Deer Occurrences

For exploring the spatial overlap of female moose, male moose, and roe deer, we first predicted the values of occurrence frequency (the effective photo/video numbers of moose or roe deer that were captured by camera trapping) for female, male moose, and roe deer during the two time periods by the optimal model simulation. Secondly, we used these predicted values in a generalized additive model (GAM) to build the relationships between female moose and male moose, female moose and roe deer, and male moose and roe deer in the two time periods. All analyses were conducted by the predict command and MGCV package in R software (Version 4.1.0).

To explore the temporal overlap of female moose, male moose, and roe deer, we first determined the respective activity times according to the time stamps recorded on camera trap photographs and videos [27]. We then separately estimated the activity patterns of female moose, male moose, and roe deer using the distribution function to run pairwise comparisons in each of the two time periods. Finally, we calculated the coefficient of overlap (Δ), which ranges from 0 (no overlap) to 1 (complete overlap). We selected Δ4 for larger sample sizes ($\geq 75$) to show the overlap index [28]. The 95% confidence intervals were calculated by 10,000 bootstrap samples, and we selected basic 0. All analyses were conducted using the "Overlap" package in R software (Version 4.1.0).

### 2.4. Temperature Effects on Female and Male Moose Population Distribution

We extracted the real time temperature when moose were captured by camera trap, and imprinted this information on each photograph or video. To explore how moose adapt to temperature, we compared real time temperatures with the threshold temperatures in moose. Specifically, a threshold temperature lower than $-5\,°C$ has been reported in winter, and greater than $14\,°C$ in summer; moose begin responding when warm temperatures are greater than $17\,°C$ in summer [29–31]. Additionally, we obtained the spatial distribution

values of monthly average temperature in the study area from the Geospatial Data Cloud Platform ([www.gscloud.cn](www.gscloud.cn)) with a grid size of 1 km. We used the "Raster Calculator" in ArcGIS 10.6 to calculate the spatial distribution values of average temperature during the two time periods. Lastly, we used a GAM to simulate the relationship between moose occurrence frequency and average temperature during the two time periods.

## 3. Results

In total, we captured 157 moose and 383 roe deer independent photos or videos by camera trapping, including 60 female moose, 54 male moose, 277 roe deer during the snow-free period, and 19 female moose, 24 male moose, and 106 roe deer during the snow period.

### 3.1. Univariate Optimal Scales of Moose and Roe Deer Occurrence

Our results showed that the scales of univariate analyses of moose and roe deer occurrence were insensitive at the small scale of 50 m and 100 m, and there were distinct differences between moose and roe deer in their respective optimal scales at which each habitat variable was of influence (Tables S2 and S3). In particular, the univariate optimal scale of moose showed aspect and distance to road were 200 m; slope and abundance of *Chosenia arbutifolia* shrub were 400 m; elevation, distance to river, abundance of *Rhododendron* shrub, distance to mixed forest, and distance to larch forest were 800 m; canopy density, herb coverage, distance to birch forest, amount of fallen log, distance to shrub swamp, abundance of *Betula exilis* shrub, and distance to forest swamp were 3200 m (Table S2). No relationship between moose occurrence and any of the habitat variables were detected at scales 50 m, 100 m, or 1600 m (Table S2).

The univariate optimal scale of roe deer showed slope, aspect, distance birch forest, distance to larch forest, and distance to mixed forest were 200 m; amount of fallen log and abundance of *Rhododendron* shrub were 800 m; abundance of *Betula exilis* shrub was 1600 m; elevation, canopy density, distance to road, distance to river, herb coverage, distance to shrub swamp, abundance of *Chosenia arbutifolia* shrub, and distance to forest swamp were 3200 m (Table S3). No relationship between roe deer occurrence and any of the habitat variables were detected at scales 50 m, 100 m, or 400 m (Table S3).

### 3.2. Habitat Selection of Female Moose, Male Moose, and Roe Deer

We obtained the most parsimonious habitat models of moose and roe deer for the whole year, and for female moose and male moose during each of the two time periods by GLM with stepwise regression (Tables S4 and S5). For moose, in the most parsimonious model for the whole year, significant habitat variables included elevation, herb coverage in a negative direction ($p < 0.05$), distance to larch forest, and abundance of *Betula exilis* shrub with a significant positive relationship ($p < 0.05$) (Table 1). Our results showed moose preferred to select habitats with lower elevation, farther distance to larch forest, higher abundance of *Betula exilis* shrubs, and lower herbaceous coverage. For roe deer, the most parsimonious model for the whole year found that the habitat variables included elevation, distance to larch forest with significant negative relationship ($p < 0.05$), slope, and abundance of *Betula exilis* shrub with significant positive relationship ($p < 0.05$) (Table 1). Our results showed that roe deer selected habitats with lower elevation, higher slopes, farther distance to larch forest, and a higher abundance of *Betula exilis* shrubs.

When dividing the year into the two time periods, our results showed that female moose in the snow-free period selected habitats with lower elevation ($p < 0.01$), lower slopes ($p < 0.01$), and farther distance to larch forest ($p < 0.01$) (Table S6). Male moose, selected habitats with lower elevation ($p < 0.01$), closer distance to mixed forest ($p < 0.05$), a higher abundance of *Betula exilis* shrubs ($p < 0.05$), and lower canopy density ($p < 0.01$) (Table S6). In the snow period, female moose selected habitats with lower elevation ($p < 0.01$), and farther distance to larch forest ($p < 0.05$) and mixed forest ($p < 0.05$) (Table S6). For male moose, they selected habitats with lower elevation ($p < 0.01$), farther distance to larch

forest ($p < 0.01$) and mixed forest ($p < 0.05$), and a higher abundance of *Betula exilis* shrubs ($p < 0.01$) (Table S6).

**Table 1.** Parameters of the most parsimonious models of GLM for moose and roe deer habitat selection during the whole year. Bold text indicates statistically significant parameters ($p < 0.05$).

| Species | Parameter | Coef. | SE | Z Value | *p* | 95% CI | |
|---|---|---|---|---|---|---|---|
| Moose | Intercept | 13.02 | 1.37 | 9.50 | <0.01 | 10.42 | 15.81 |
| | Elevation (800 m) | −0.01 | 0.00 | −8.71 | <0.01 | −0.02 | −0.01 |
| | Distance to larch forest (800 m) | 0.00 | 0.00 | 3.58 | <0.00 | 0.00 | 0.00 |
| | Abundance of *Betula exilis* shrub (3200 m) | 0.51 | 0.12 | 4.31 | <0.01 | 0.28 | 0.74 |
| | Herb coverage (3200 m) | −0.75 | 0.30 | −2.49 | <0.05 | −1.34 | −0.15 |
| Roe deer | Intercept | 4.80 | 0.54 | 8.82 | <0.01 | 3.74 | 5.88 |
| | Elevation (3200 m) | −0.00 | 0.00 | −7.13 | <0.01 | −0.01 | −0.00 |
| | Slope (200 m) | 0.06 | 0.01 | 4.24 | <0.01 | 0.03 | 0.09 |
| | Distance to larch forest (200 m) | 0.00 | 0.00 | 4.28 | <0.01 | 0.00 | 0.00 |
| | Abundance of *Betula exilis* shrub (1600 m) | −0.32 | 0.13 | −2.52 | <0.05 | −0.57 | −0.08 |

When comparing the contribution rate of overlapping variables in the optimal models of female and male moose in the two time periods, our results showed that the contribution rate of elevation for female and male moose models during the snow-free period was higher than during the snow period (Figure 2a). The contribution rate of far away larch forest of the male moose model in the snow season was higher than for female moose (Figure 2b), while female moose selected areas that were nearer to larch forest in the snow-free period than in the snow period. Male and female moose also had opposite selection characteristics with regards to mixed forest (Figure 2c). The contribution rates of abundance of *B. exilis* shrubs were similar among female and male moose during the two time periods, though we note that female moose did not show a clear relationship with *Betula exilis* shrub abundance during the snow-free period (Figure 2d).

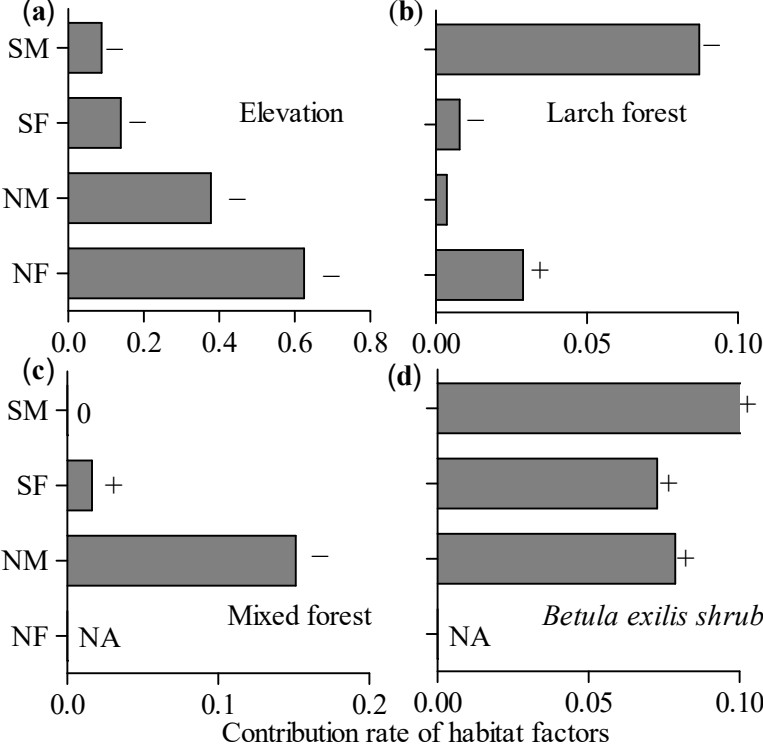

**Figure 2.** Contribution rate of female and male moose habitat parameters in the most parsimonious univariate models during the snow-free and snow seasons, including elevation (**a**), larch forest (**b**),

mixed forest (**c**) and *Betula exilis* shrub (**d**). Males in the snow season (SM), females in the snow season (SF), males in the snow-free season (NM), females in the snow-free season (NF). "+" indicates a positive relationship, "−" indicates a negative relationship, and no symbol indicates no significant relationship, "NA indicates not applicable".

### 3.3. Spatial Overlap of Female Moose, Male Moose, and Roe Deer

We used GAM to explore the spatial overlap of female moose, male moose, and roe deer during the two seasons. Our findings showed that, spatially, female moose had a significant positive relationship with male moose ($p < 0.01$) during the two time periods (Figure 3a,b). Male moose had a significant positive relationship ($p < 0.01$) with roe deer during the two periods (Figure 3c,d); female moose also had a significant positive relationship ($p < 0.01$) with roe deer during the two periods (Figure 3e,f). By modelling the occurrence frequency of moose and roe deer, our results strongly supported the spatial overlap of female moose, male moose, and roe deer, and their spatial overlaps were independent of sex or season.

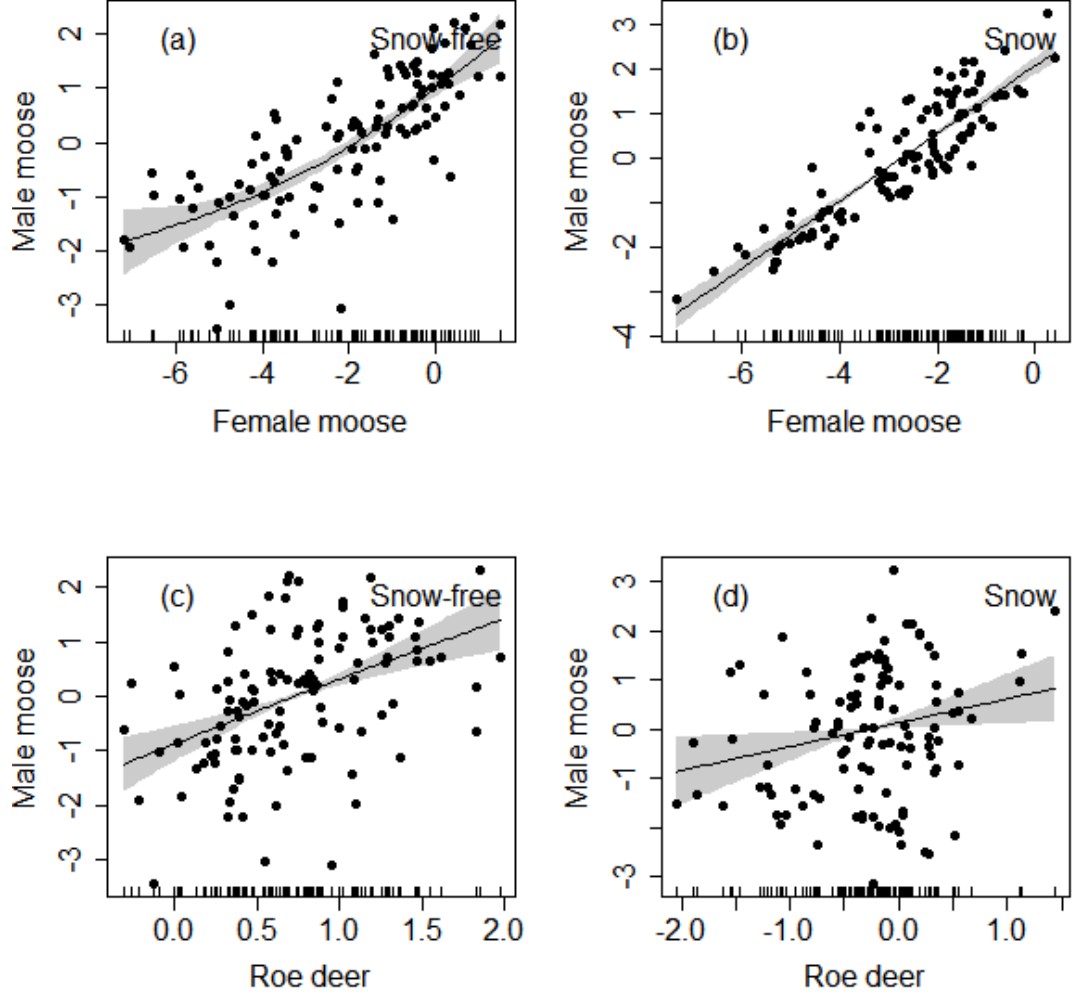

**Figure 3.** *Cont.*

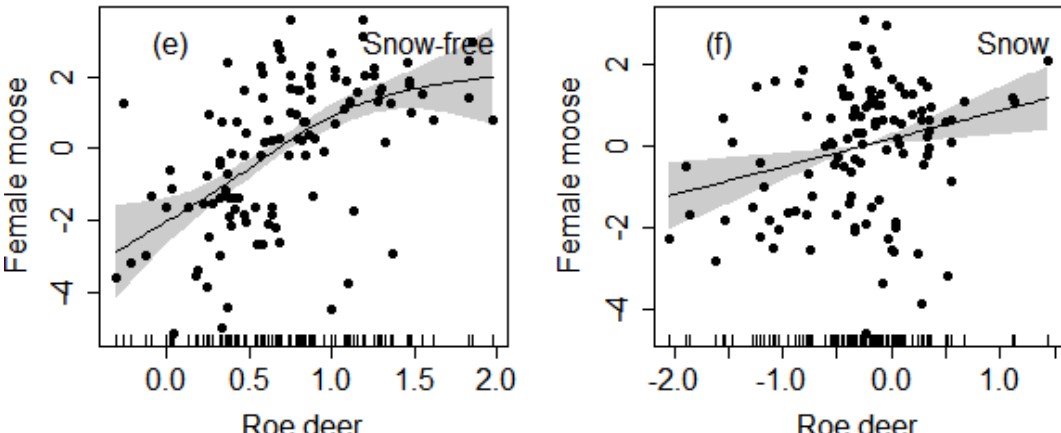

**Figure 3.** Spatial overlap relationships of female and male moose with roe deer created by modelling the occurrence frequency during the snow−free and snow time periods/seasons; including the spatial overlap of male moose and female moose during snow−free (**a**) and snow periods (**b**), male moose and roe deer during snow−free (**c**) and snow periods (**d**), female moose and roe deer during snow−free (**e**) and snow periods (**f**).

### 3.4. Temporal Overlap of Female Moose, Male Moose, and Roe Deer

The activity patterns of female moose and male moose were more active at dawn and dusk during the snow-free period, while roe deer were more active at dawn during this snow-free period; during the snow period, female moose and male moose were more active at daytime, while roe deer were more active before dusk (Figure 4). In general, the female moose activity patterns almost synchronized with those of male moose during the snow-free period ($\Delta = 0.87$, 95% CI 0.77–0.98) (Figure 4a). However, their activity patterns appeared to have less synchronization during the snow period, regardless of the observed high overlap index ($\Delta = 0.79$, 95% CI 0.60–0.97) (Figure 4b). Regarding inter-specific temporal overlap, our results from the snow-free period showed that the $\Delta$ of male moose with roe deer was 0.69 (95% CI 0.58–0.80) (Figure 4c), and the $\Delta$ of female moose with roe deer was 0.65 (95% CI 0.54–0.75) (Figure 4e); during the snow period, the $\Delta$ of male moose with roe deer was 0.69 (95% CI 0.53–0.84) (Figure 4d), and the $\Delta$ of female moose with roe deer was 0.69 (95% CI 0.54–0.85) (Figure 4f). Our findings confirmed that the female moose and male moose activity patterns showed a degree of temporal separation with roe deer during both time periods.

### 3.5. Relationships between Temperature and Moose Distribution

We obtained the real time temperature of moose occurrence points by camera trapping monitoring at each month, and compared the temperatures at these points with moose threshold temperatures, which referred to the published literature [29–31]. Our results found that the temperatures of the moose distribution points in this study were lower than 17 °C, especially from April to September, which showed that moose selected the areas with suitable temperatures for resisting heat stress during the snow-free period (Figure 5). The temperature of the moose distribution points from November to the following March were lower than −5 °C, which showed that moose facing cold environments withstood cold temperatures during the snow period (Figure 5).

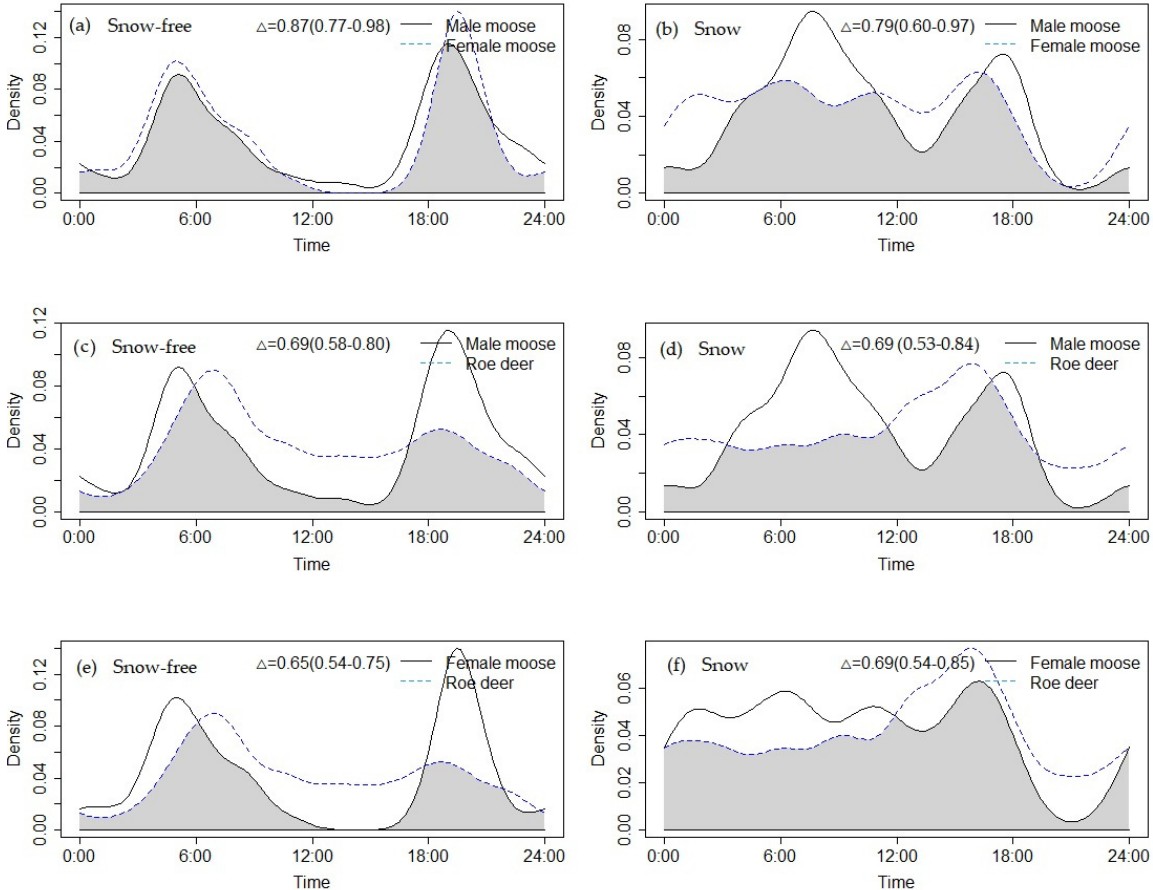

**Figure 4.** Temporal overlap of daily activity patterns of female and male moose during the snow−free and snow time periods/seasons, and the respective overlap with roe deer during both time periods; including the temporal overlap of male moose and female moose during snow−free (**a**) and snow periods (**b**), male moose and roe deer during snow−free (**c**) and snow periods (**d**), female moose and roe deer during snow−free (**e**) and snow periods (**f**).

We used GAM to explore the relationships between moose encounter frequency (the effective photo/video numbers of moose or roe deer were captured by camera trapping) with the average environmental temperature during the two time periods. Our results found that both female and male moose exhibited a significant negative relationship with temperature during the snow-free period ($p < 0.05$) (Figure 6a,c), which confirmed that moose selected the areas with suitable temperatures during this period, and high temperature areas were the least visited. During the snow period, we did not detect any significant relationships between either female or male moose encounter frequency and temperature ($p > 0.05$) (Figure 6b,d), which showed that moose distribution was not affected by temperature during this period, which might be in relation to the small sample sizes of both female and male moose.

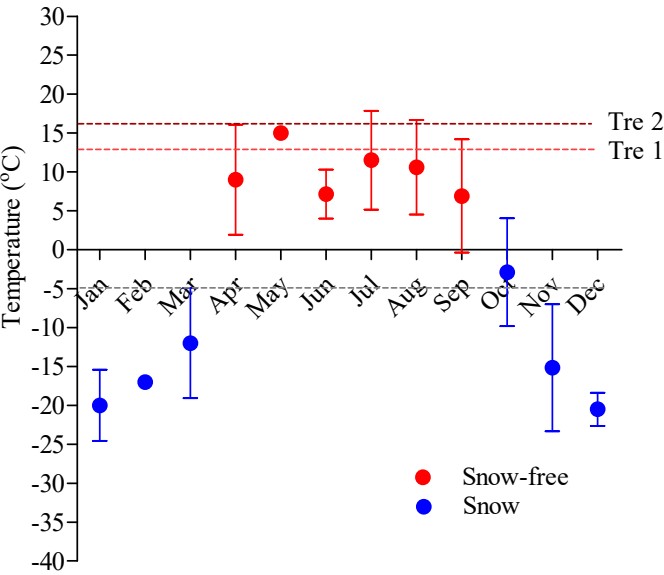

**Figure 5.** Temperature of moose occurrence points in each month, showing that moose occurred at temperatures above the threshold of 14 °C (Tre 1) during summer (during snow−free period), and that moose occurrence did not transgress the upper threshold of 17 °C (Tre 2) during summer (snow−free period). The gray dotted line showed moose occurred at temperatures lower than the −5 °C in winter (during snow period).

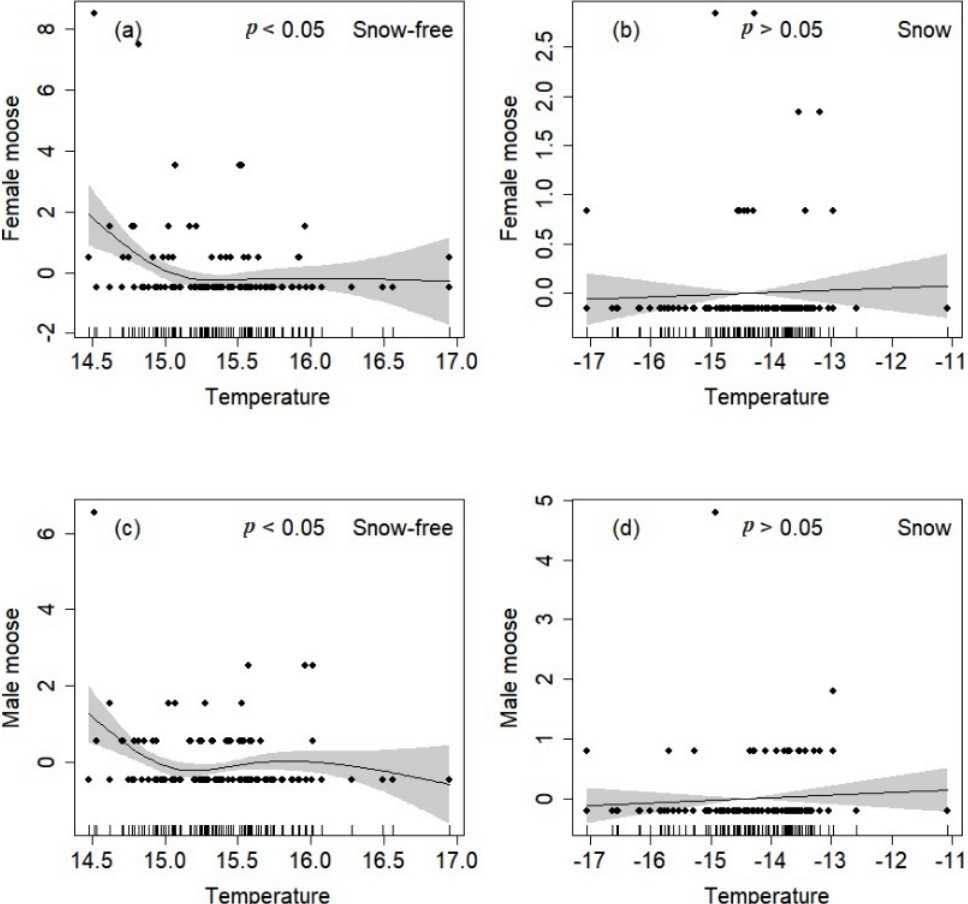

**Figure 6.** Relationships between female (**a,c**) and male (**b,d**) moose encounter frequency and local temperature (°C) at camera trap points during the snow−free and snow time periods/seasons.

## 4. Discussion

### 4.1. Effect of Scale on Moose and Roe Deer Habitat Selection

Scale is an important factor in habitat selection, because some aspects of wildlife ecology are scale-specific [32]. The selection of inappropriate scales during analyses, or even of a single scale, may obtain incomplete or inaccurate habitat selection results [33,34]. Scales of large ungulates range from feeding or browsing sites to home range and even landscape scale, which mainly occur at larger scales because they need to account for food selection and predation pressure [35,36].

Moose and roe deer have a similar pattern of habitat selection in the Greater and Lesser Khingan Mountains in China, but they vary in their feeding sites at landscape, patch, and microhabitat scales [11]. Our results showed that moose and roe deer spatial use did not respond to the individual variables investigated here at fine scales, especially at 50 m and 100 m resolutions. Rather, the scales at which the two herbivores appeared to select areas were related to those of their resources requirements, while there existed variation in the optimal scale for each habitat variable. In particular, we found that the habitat selection of moose was negatively related to the optimal scale of elevation at 800 m, and the habitat selection of roe deer was negatively related to the optimal scale of elevation at 3200 m. The habitat selection of moose was positively related to the optimal scale of distance to larch forest at 800 m, and that of the roe deer was positively related to the optimal scale of distance to larch forest at 200 m. The optimal scale of *Betula exilis* shrub abundance for the significant habitat relationships of moose and roe deer was 3200 m and 1600 m, respectively. Notably, moose selected habitats with greater abundances of *Betula exilis* shrubs, unlike roe deer.

This advances the understanding gained by many other studies, which have proven that the variation in scale selection in moose habitat selection studies may result in different selection patterns being observed or interpreted [17,37–39]. The moose selection mechanism of univariate selection with different scales should be discussed in depth, such as whether it was driven by food resources or by specific behaviors (reproductive behavior, etc.) or not. In addition, the comprehensive effects of scale on the spatial competition between moose and their competitive species are still lacking. Therefore, in the future, we not only need to consider the driving mechanism of univariate optimal scale selection on the modelling on the single species, but also need to consider the scalar effects on the spatial competition patterns of multi-species.

### 4.2. Differences in Female and Male Moose Habitat Selection

The habitat selection of wildlife needs to consider numerous factors, such as food quality and quantity, cover condition, and predation risk [40]. These trade-offs are often related to the variation in temporal and spatial habitat conditions, as well as sex and season [41].

Moose move to higher elevations with leeward slopes which face the sun, as greater snow depths reduce their relative frequency in the valley during the winter [42]; our findings confirmed that moose preferred to select higher elevations with leeward slopes during the snow period than during the snow-free period. Female and male deer select different habitats outside of the mating season, because of the variation in nutritional requirements [43] and physiological needs (such as gestation and lactation) [44] related to life-history features, as well as the avoidance of predation pressure [15]. Our results also showed that female and male moose selected lower elevation habitats more so during the snow-free period than during the snow period. It has also been reported that male moose select habitats with higher food quantity, while female moose select habitats with higher food quality, because of different nutritional requirements at specific periods [17]. Here, our findings from the snow-free period showed that male moose preferred to select habitats with a greater abundance of *Betula exilis* shrubs, while female moose did not demonstrate this relationship. However, during the snow period, both male and female moose selected habitats with greater abundances of *Betula exilis* shrubs; it is possible that severe winter weather conditions caused the concentrated feeding on specific foods.

### 4.3. Effect of Roe Deer Occurrences on Moose Distribution and Daily Activity

Resource competition between moose and roe deer occurred in a three-dimensional ecological niche—namely, food type, feeding diameter/height, and habitat [11]. Our study also found that the spatial distribution of moose and roe deer overlapped during the two time periods, but there were differences in the scales at which variables were evidently optimal and hence applicable in the multi-scale optimal habitat selection model. The multivariate model considering the univariate optimal scale could contribute to understanding the spatial competition/overlap between moose and roe deer for selection of the same habitat, and could be adjusted by different selection scales of the same habitat factor. Our results found the optimal scale for slope and moose was 400 m compared to 200 m for roe deer; this meant that moose moved around more. Our findings also confirmed that moose preferred lower altitude habitats more than roe deer, which relates to the physical characteristics of the two species and the landscape structure, with more swamp in the low-altitude area being more suitable for moose as a low-altitude habitat than for roe deer, as moose legs are longer than those of roe deer. The results showed that moose are better able to utilize swamp habitats than roe deer. Additionally, roe deer selected the larch forest less than did moose, which might relate to the high proportion of larch forest with a low availability of food in the study area.

Temporally, competitive species should stagger daily activity times. We have shown that moose and roe deer have high spatial overlap, and our findings showed that the activity patterns of both sexes of moose showed a degree of temporal separation with roe deer during both time periods. Such spatial and temporal interaction patterns might be one of the key factors allowing the long-term coexistence of moose and roe deer. Specifically, during the snow-free period, roe deer were more active at daybreak, while moose were more active at dawn and nightfall; during the snow period, roe deer were more active before nightfall, while moose were more active at daytime. These relatively staggered peak activity patterns might weaken direct competition for food resources by the two species. In addition, our results also showed the seasonal change of roe deer daily activity patterns, in which the peak was from daybreak to noon during the snow-free period and from noon to evening during the snow period; these results indicated that seasonal weather changed roe deer activity patterns [41,45]. Harsh weather conditions might also trigger ecological overlap for roe deer and fallow deer (*Dama dama*) [46]. Although our results showed a weak—but higher—overlap index between moose and roe deer during the snow period, this might also have been affected by the shortage of food resources during the harsh winter.

### 4.4. Effect of Temperature on Moose Distribution and Daily Activity Patterns

Endotherms regulate body temperature by behavioral and physiological responses in order to adapt to the change in ambient temperature, and the external environmental temperature sometimes exceeds their tolerance. The threshold temperatures of moose have been found to be higher than 14 °C in summer, during which moose begin responding when warm temperatures rise above 17 °C, which has became synonymous with the upper critical temperature for moose [47–51]. Our results showed that the temperatures of moose activity areas were lower than 17 °C, especially from April to September during the snow-free period, which further validated the value of this threshold.

Daily activity times can also vary between sexes, as demonstrated, for example, in white-tailed deer (*Odocoileus virginianus*), and furthermore, seasonal weather can change activity patterns [41,45]. However, our findings showed that the activity patterns of female moose and male moose were well synchronized during both time periods, though their activity patterns appeared to have less synchronization—regardless of high overlap index—during the snow period than during the snow-free period. We conclude that the variation of activity patterns might be driven by the change in seasonal environmental conditions, such as temperature, habitat, and food sources.

Ambient temperature might influence an animals' activity synchronization [52], and deer activity mainly occurs within a certain temperature range [41]. If ambient temperatures exceed the moose tolerance threshold, they will reduce food intake and increase energy intake because of the decreased digestibility of dry matter, and heat stress will impair their ability to take in adequate energy and nutrients [53,54]. As mentioned above, moose preferred habitats below the threshold temperature values. With the increase in ambient temperature during the summer, moose selected activity time in the afternoon and habitats with high canopies to avoid the periods of high heat stress [55]. Our results showed that both sexes of moose were more active at dawn and nightfall in order to avoid the high temperature in the daytime during the snow-free period; these might relate to their lower heat tolerance [20,21,53,54].

Our published paper showed that the late spring temperature increased by 3.01C from 1969 to 2009 in this area [24]. Moose belong to climate-sensitive ungulate species, and many studies showed that they adapted to cold well and that are cold adapted [20,56,57]. Moose activity did not significantly change with the increase in ambient temperature during the winter [22,58]; this result might also be in relation to the small sample sizes of both the female and male moose. Our findings also showed that the temperatures in moose activity areas were lower than $-5\,^{\circ}\text{C}$ from November to the following March (i.e., during the snow period). Moose further change their activity patterns and space use degrees in order to avoid the change in temperature and to obtain sufficient food during different seasons [59]. In particular, our findings showed that the daily activity of female and male moose increased at daytime, which might be the strategy of cold adaption for moose during the snow period because of warmer environmental temperatures during the daytime due to sunshine; this activity pattern would also reduce energy consumption due to very low temperatures at nighttime.

## 5. Conclusions

In this research, we first established that moose and roe deer select habitats at different spatial scales, and that sex and season can help explain differences in moose habitat selection. Secondly, although these competitive species of moose and roe deer have high spatial overlap, they have a degree of temporal separation, and, especially, their daily activity patterns indicate seasonal change. Lastly, we found that moose selected the habitats within a certain temperature range below $17\,^{\circ}\text{C}$, which is evidently a significant temperature driving moose population distribution and daily activity patterns during the snow-free period. This study provides new insights into how the comprehensive effects of scale, temperature, and inter and intra-specific relationships influence moose habitat selection and daily activity patterns, and have scientific implications for the protection and management of large, heat-sensitive species during an era of sustained global warming.

**Supplementary Materials:** The following supporting information can be downloaded at: https://www.mdpi.com/article/10.3390/rs14246401/s1, Table S1. Description of main habitat factors in Hanma National Nature Reserve used in multi-scale analyses; Table S2. Optimal scale of habitat variables influencing moose occurrence, and respective parameter values, determined by generalized linear model (GLM) prediction; Table S3. Optimal scale of habitat variables influencing roe deer occurrence, and respective parameter values, determined by generalized linear model (GLM) prediction; Table S4. Number of model parameters (K), Akaike's Information Criterion (AIC) scores and $R^2$ of the most supported GLM (family = binomial) by stepwise regression for moose and roe deer resource selection; Table S5. Number of model parameters (K), Akaike's Information Criterion (AIC) scores and $R^2$ of the most supported GLM (family = binomial) by stepwise regression for female and male moose resource selection in the snow-free and snow period; Table S6. Parameter values of the most parsimonious models of general linear models (GLM) for female and male moose occurrence frequency in the snow-free and snow periods.

**Author Contributions:** G.J. designed the study. H.B., P.Z., D.W., W.Z., Y.L., X.L., F.Y. (Feifei Yang) F.Y. (Fan Yang) contributed to field survey. N.J.R. contributed to the review and editing of the manuscript.

H.B., Y.X. and G.J. contributed to data analysis, paper writing and revising. All authors have read and agreed to the published version of the manuscript.

**Funding:** This research was supported by National Natural Science Foundation of China (NSFC31872241, 32100392); the Heilongjiang Postdoctoral Foundation (LBH-Z20108); and the Fundamental Research Funds for the Central Universities (2572022DS04).

**Data Availability Statement:** Not applicable.

**Conflicts of Interest:** The authors declare no conflict of interest.

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
