# Peer review of "Effects of Inter- and Intra-Specific Interactions on Moose Habitat Selection Limited by Temperature"

_remotesensing, doi:10.3390/rs14246401_

Round 1

Reviewer 1 Report

Line 143: Change follow to following

Line 183: There is a lot going on in Tables S2 & S3 and a notable difference in the optimal scales for moose vs. roe deer. I was confused about what this result means – For example, the optimal scale for slope and moose is 400 m compared to 200 m for deer. Can you explain this more in the Discussion? Does it just mean moose move around more?

Line 211: Indicate in the Table heading that this is for the whole year.

Fig 4: There are problems with the legend’s position – you may need to move the text up.

Fig 5: Some Figures are called Fig. and other Figures – please standardise.

Fig 5: Shouldn’t -5 C also be labelled as a winter threshold?

Line 289: Add “during” before the snow period.

Line 290: Remove extra r.

Fig: Not a lot of data for moose in the snow period, particularly for female moose – this should be commented on in the Discission. How could sample size affect the results?

Line 405: Change in to “to”.

Discussion: Ok – you have a snap-shot in time – what could the implications of global warming be for moose, and how much have temperatures changed in this area, say, over the past 20 years? You then highlight the impacts of ambient temperature on daily activity – what should researchers focus on next to confirm this as a big problem? 

Author Response

Dear Editor and reviewers,

Thank you so much for the major revision to resubmit our manuscript (remotesensing-1996844). According to the referees’ and editor’s recommendations and comments, we had modified the manuscript.

  In order to let you have a better understanding of this manuscript, we have provided a version of the revision based on the three reviewers. We hope our revision is satisfactory to you. However, we are happy to consider any further requests you or referees may make. Please see the attachment with the point-by-point responses to the your comments.

Thank you for your time considering our revised manuscript for publication in Remote sensing.

Best Regards,

Ref: remotesensing-1996844

Title: Effects of Inter- and Intra-specific Interactions on Moose Habitat Selection Limited by Temperature

Guangshun Jiang

On behalf of all co-authors

Reviewer 2 Report

This manuscript was made harder to read by the poor English expression throughout - which distracted from the core content. Rather than provide an itemised list of corrections to the language here, please see the annotated version of your manuscript that I have provided. This is not an exhaustive attempt to rectify the poor writing and I encourage you to not only tend the edits I have indicated but also look thoroughly elsewhere across the text.

More generally, I have also provided comments throughout which require you to provide further detail about your work and, importantly, how it differs from previously published and related literature. This is not entirely clear and made worse by generalised statements - rather than pointing out exactly how your work contributes and/or can be set apart. Once again, see the annotated manuscript.

Aside from that, further statement needs to be made about the very low sample sizes you have used to infer habitat differences between moose and roe deer, particularly for the former species in the snow season. This is reflected in part by the large confidence intervals about estimates, including activity periods. 

In the same vein, further justification needs to be given about the habitat variables you measured, and the different scales at which you measured them. Why were they important? Without proper justification the scale at which you measured attributes seems a bit arbitrary. Perhaps the inclusion of information about the home range sizes of moose and deer would help matters?

Using data from camera traps to infer "real temperature" is fraught. What can you say about how accurate the temperature sensors in cameras are? At best they are capable of determining gross differences. This needs to be stated in the text.

The implications of your results for management of moose and roe deer also deserve to be articulated. In the conclusion you state the findings are important for that purpose, but don't detail why. For example, if there are certain positive relationships with either species and a certain habitat type, does that mean special effort needs to be made to preserve that habitat type? Or is it protected anyway? And in relation to global warming, what do your findings say? Your study wasn't designed to address that issue directly, so more detail is needed.

Author Response

(The authors gave the same response as above.)

Reviewer 3 Report

I did some yellow markings in the text regarding some redactions or words that could be improved. See attached text.

Author Response

(The authors gave the same response as above.)

Round 2

Reviewer 2 Report

Thank you for tending the suggested edits in full.